# Deep convolutional neural networks to predict cardiovascular risk from computed tomography

Roman Zeleznik [1,2,3], Borek Foldyna[1,2], Parastou Eslami[2], Jakob Weiss[1,2,3,4], Ivanov Alexander[2], Jana Taron[2,4], Chintan Parmar [1,3], Raza M. Alvi[2], Dahlia Banerji[2], Mio Uno[2], Yasuka Kikuchi[2,5], Julia Karady[2,6], Lili Zhang[2], Jan-Erik Scholtz[2], Thomas Mayrhofer [2,7], Asya Lyass[8], Taylor F. Mahoney [9], Joseph M. Massaro[9], Ramachandran S. Vasan [10,11], Pamela S. Douglas[12], Udo Hoffmann[1,2,15], Michael T. Lu [1,2,15] & Hugo J. W. L. Aerts [1,2,3,13,14,15 ✉]

Coronary artery calcium is an accurate predictor of cardiovascular events. While it is visible on all computed tomography (CT) scans of the chest, this information is not routinely quantified as it requires expertise, time, and specialized equipment. Here, we show a robust and time-efficient deep learning system to automatically quantify coronary calcium on routine cardiac-gated and non-gated CT. As we evaluate in 20,084 individuals from distinct asymptomatic (Framingham Heart Study, NLST) and stable and acute chest pain (PROMISE, ROMICAT-II) cohorts, the automated score is a strong predictor of cardiovascular events, independent of risk factors (multivariable-adjusted hazard ratios up to 4.3), shows high correlation with manual quantification, and robust test-retest reliability. Our results demonstrate the clinical value of a deep learning system for the automated prediction of cardiovascular events. Implementation into clinical practice would address the unmet need of automating proven imaging biomarkers to guide management and improve population health.

[1] Artificial Intelligence in Medicine (AIM) Program, Mass General Brigham, Harvard Medical School, Boston, MA, USA. [2] Cardiovascular Imaging Research Center, Massachusetts General Hospital, Harvard Medical School, Boston, MA, USA. [3] Department of Radiation Oncology, Brigham and Women's Hospital, Dana-Farber Cancer Institute, Harvard Medical School, Boston, MA, USA. [4] Department of Diagnostic and Interventional Radiology, Eberhard Karls University of Tübingen, Tübingen, Germany. [5] Center for Cause of Death Investigation, Faculty of Medicine, Hokkaido University, Sapporo, Hokkaido, Japan. [6] Cardiovascular Imaging Research Group, Heart and Vascular Center, Semmelweis University, Budapest, Hungary. [7] School of Business Studies, Stralsund University of Applied Sciences, Stralsund, Germany. [8] Department of Mathematics and Statistics, Boston University, Boston, MA, USA. [9] Department of Biostatistics, Boston University School of Public Health, Boston, MA, USA. [10] National Heart, Lung, and Blood Institute and Boston University, Framingham Heart Study, Framingham, MA, USA. [11] Departments of Cardiology and Preventive Medicine, Department of Medicine, Boston University School of Medicine, Boston, MA, USA. [12] Department of Medicine, Division of Cardiology, Duke University School of Medicine, Duke Clinical Research Institute, Durham, NC, USA. [13] Department of Radiology, Brigham and Women's Hospital, Dana-Farber Cancer Institute, Harvard Medical School, Boston, MA, USA. [14] Radiology and Nuclear Medicine, CARIM & GROW, Maastricht University, Maastricht, The Netherlands. [15]These authors jointly supervised this work: Udo Hoffmann, Michael T. Lu, Hugo J.W.L. Aerts. ✉email: haerts@bwh.harvard.edu

Cardiovascular disease is the most common preventable cause of death, accounting for up to 45% of the mortality in Europe[1] and 31% in the United States[2]. Effective lifestyle and pharmacological prevention is available, but identifying those who would benefit most remains an ongoing challenge[3]. Traditional risk factors, such as age and sex, have limited accuracy for predicting cardiovascular disease among individuals. Hence, efforts are needed to further improve cardiovascular risk prediction and stratification on an individual basis[4].

One of the strongest known predictors for adverse cardiovascular events is coronary artery calcification, which can be quantified on computed tomography (CT)[5,6]. The CT coronary calcium score is a measure of the burden of coronary atherosclerosis and is one of the most widely accepted measures of cardiovascular risk[5,6]. Coronary calcium scoring has been recommended by the guidelines for risk stratification, specifically in the setting of primary prevention in asymptomatic individuals[7,8]. In symptomatic patients, the presence of coronary calcium is associated with future cardiovascular events in the stable chest pain setting[9] and low likelihood of acute coronary syndrome in patients with acute chest pain[10]. Additionally, showing patients their coronary calcium provides a "teachable moment" to empower them to make informed, individualized decisions, and to improve long-term compliance for preventative therapy and lifestyle changes including smoking cessation[11,12].

While the calcium score has been traditionally measured on specialized electrocardiography (ECG)-gated cardiac CT, it can also be measured on nearly every standard CT scan of the chest performed without contrast[8]. However, the measurement requires radiological expertise, time, and specialized coronary calcium quantification equipment. As a result, this essentially free available information is usually not reported. An automated system for quantifying calcium on medical imaging could help put this actionable information into the hands of patients and their physicians.

Recent strides in artificial intelligence, deep learning in particular, have shown its viability in several medical applications such as medical diagnostic and imaging, risk management, or virtual assistants. Especially in medical imaging there is a large potential as deep learning can successfully be used for identifying and segmenting objects within the 3-dimensional (3D) image space[13–16]. A major advantage is that deep learning can automate complex assessments that previously could only be done by radiologists, but now is feasible at a scale with a higher speed and lower cost. This makes deep learning a promising technology for automating cardiovascular event prediction from imaging. Before clinical introduction can be considered, however, the generalizability of these systems needs to be demonstrated as they need to be able to predict cardiovascular events of asymptomatic and symptomatic individuals across multiple clinical scenarios, and work robustly on data from multiple institutions.

Here, we present a deep learning system that automatically and accurately can predict cardiovascular events by quantifying the presence and extent of coronary calcium. The system was evaluated in 20,084 individuals from four well-established prospective cohorts and randomized controlled trials—a healthy asymptomatic community-dwelling sample from the Framingham Heart Study (FHS)[17], older asymptomatic heavy smokers in the National Lung Screening Trial (NLST)[18], a symptomatic stable chest pain population evaluated for suspected coronary artery disease in the outpatient setting in the Prospective Multicenter Imaging Study for Evaluation of Chest Pain (PROMISE)[19], and a symptomatic acute chest pain population presenting to the emergency department in the Rule Out Myocardial Infarction using Computer Assisted Tomography (ROMICAT-II)[20] trial. Overall, the association between the algorithm's prediction and

adverse cardiovascular events was tested in individuals who were imaged using different CT scanners, applying a variety of CT scan protocols, including ECG-gated and non-gated CT scans. Accuracy compared to the gold standard of expert human readers was assessed in 5521 subjects across all four cohorts. Our results demonstrate that deep learning methods can automate cardiovascular risk predictions from medical images acquired in several clinical scenarios. These observations provide a rationale to implement this technique in both screening and hospital settings to improve population health, at high speed and low costs.

## Results

We developed a deep learning system to automatically identify individuals at high risk for cardiovascular disease and tested the system's performance in four large independent held-out cohorts with a variety of clinical presentations and CT scanning techniques. Figure 1 provides an overview of the test cohorts and analyses. Clinical characteristics of the test cohorts can be found in Table 1.

**Development of the deep learning system**. The FHS is a long-term cardiovascular cohort study including asymptomatic persons originally from the city of Framingham in Massachusetts[17,21]. The Offspring and Third Generation FHS cohorts received ECG-gated non-contrast cardiac CT and were included in our analysis. We developed the deep learning system in the first cohort of FHS participants to have cardiac CT (FHS-CT1), including 1636 individuals. The deep learning system was trained to identify and quantify coronary artery calcium (CAC) based on manual segmentations performed by expert CT readers (Fig. 1a). To localize and segment the heart in a given CT scan, two consecutive deep learning networks were trained using 129 cardiac ECG-gated CTs with volumetric heart segmentations provided by expert readers. These networks were tested in an independent subset of our test cohorts including 1857 cardiac gated and low-dose chest screening CT (Supplementary Fig. 2a). In this test cohort the heart localization network was able to predict the heart center with an accuracy of $9 \pm 7$ mm, while the heart segmentation network achieved a Dice coefficient of $0.90 \pm 0.059$. Supplementary Table 4 provides details about the results of the two networks in the sub-cohorts. Next, the system automatically identified and segmented the coronary calcium and computed the CAC scores, and stratified them into clinically relevant categories of very low (CAC = 0), low (CAC = 1–100), moderate (CAC = 101–300), and high (CAC > 300). The system could analyze each image on an average of 1.938 s per scan on a graphics processing unit (GPU) system. Resulting CAC scores were evaluated in the test cohorts in terms of agreement with expert readers as well as predicting the risk of cardiovascular events on follow-up (Fig. 1b).

**Automated coronary calcium scoring in lung cancer screening**. To evaluate the value of coronary calcium in heavy smokers having lung cancer screening CT, we applied our deep learning system to 14,959 participants in the low-dose chest CT arm of the NLST. NLST low-dose chest CT was performed at 33 institutions with a variety of CT scanners using a non ECG-gated low-dose chest CT protocol[18].

We investigated the association between our deep learning system's calcium score and incident atherosclerotic cardiovascular disease (ASCVD) death in lung cancer screening eligible individuals with a median follow-up time of 6.7 years. Kaplan–Meier analysis and Cox regression showed significant differences between all four calcium risk groups (Fig. 2a). Adjusted for age, sex, diabetes, heart disease, hypertension, and

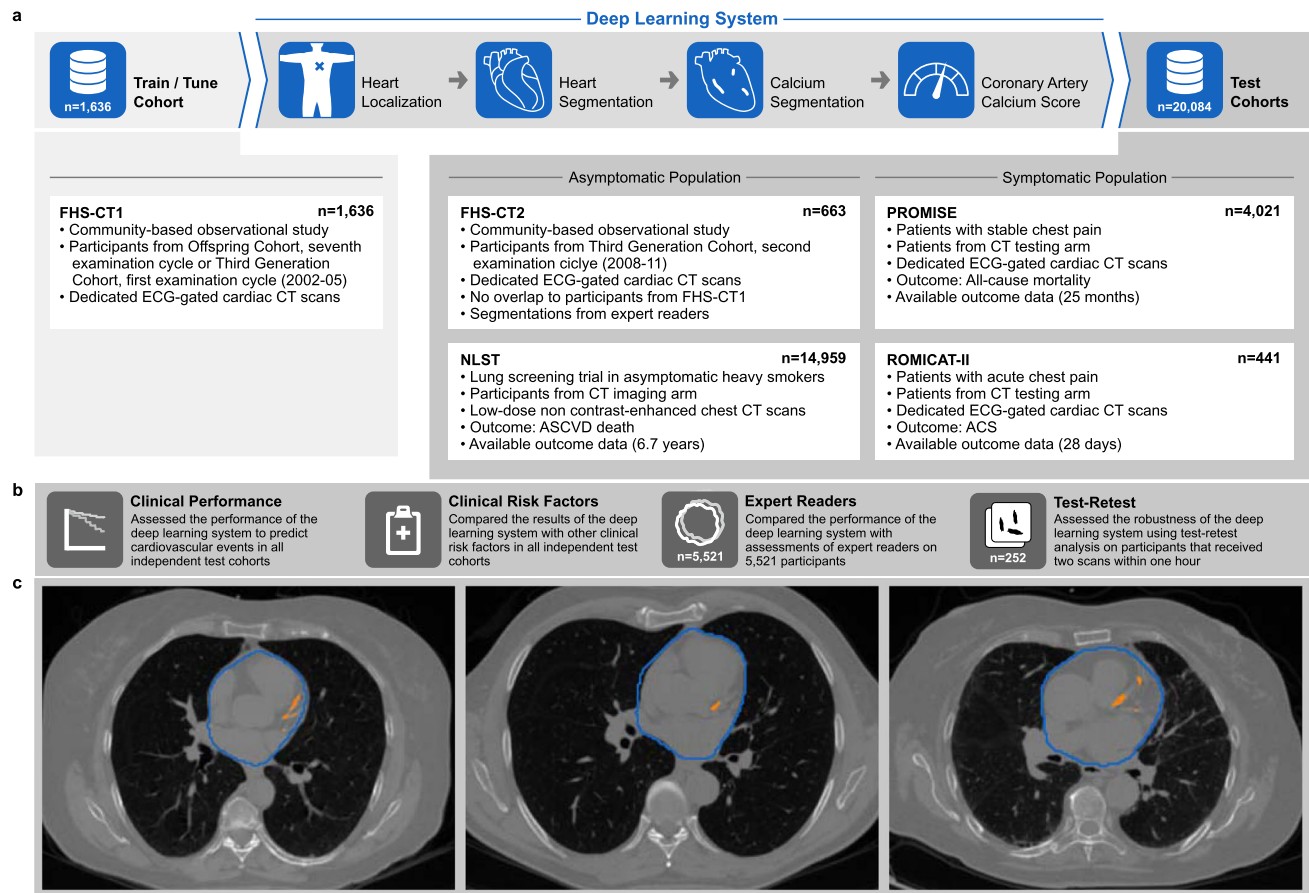

**Fig. 1 Overview of the deep-learning framework, the training and test cohorts, and the implemented evaluation steps. a** The deep-learning framework was trained and tuned on 1636 computed tomography (CT) scans from Framingham Heart Study (FHS)-CT1. In four consecutive steps a coronary calcium risk score was calculated in a fully automatic fashion. Independent testing was performed on 20,084 CT scans from four different clinical cohorts. **b** The performance of the framework was evaluated with respect to its clinical value and robustness. **c** CT scans of three representative patients of FHS-CT2 outlined with the deep learning system heart (blue contours) and coronary calcium (orange contours). FHS-CT1[17], FHS-CT2[17] Framingham Heart Study, (CT1) participants from the seventh examination cycle of the Offspring Cohort or first examination cycle of the Third Generation Cohort (2002–05) and (CT2) participants from the second examination cycle of the Third Generation Cohort (2008–11), NLST[18] National Lung Screening Trial, PROMISE[19] Prospective Multicenter Imaging Study for Evaluation of Chest Pain, ROMICAT-II[20] Rule out Myocardial Infarction using Computer Assisted Tomography II, ECG electrocardiographic, CT computed tomography, ASCVD atherosclerotic cardiovascular disease, ACS acute coronary syndrome.

**Table 1 Baseline characteristics of subjects in the four test cohorts.**

| Characteristics[a] | FHS-CT2 (n = 663) | NLST (n = 14,959) | PROMISE (n = 4021) | ROMICAT-II (n = 441) |
|---|---|---|---|---|
| Woman—n (%) | 372 (56.1) | 6,110 (40.9) | 2,047 (50.9) | 235 (53.3) |
| Age - years | 57.2 ± 11.4 | 61.5 ± 5.1 | 60.6 ± 8.0 | 53.7 ± 8.0 |
| Body mass index—kg/m$^2$ | 28.6 ± 5.5 | 27.9 ± 5.1 | 30.4 ± 5.9 | 29.3 ± 5.2 |
| Arterial hypertension—n (%) | 219 (33.1) | 5321 (35.6) | 2614 (65.0) | 233 (52.8) |
| Diabetes—n (%) | 31 (4.87) | 1427 (9.5) | 838 (20.8) | 74 (16.8) |
| Hypercholesterolemia—n (%) | 236 (35.6) | n/a | 2734 (68.0) | 198 (44.9) |
| Former or current smoker—n (%) | 202 (31.7) | 14,959 (100) | 2078 (51.7) | 220 (49.9) |
| Framingham risk score | 0.10 ± 0.1 | n/a | 0.22 ± 0.2 | n/a |
| TIMI risk score | n/a | n/a | n/a | 0.13 ± 0.3 |

FHS-CT2[17] Framingham Heart Study, participants from the second examination cycle of the Third Generation Cohort, NLST[18] National Lung Screening Trial, PROMISE[19] Prospective Multicenter Imaging Study for Evaluation of Chest Pain, ROMICAT-II[20] Rule Out Myocardial Infarction using Computer Assisted Tomography II, TIMI thrombolysis in myocardial infarction. n/a: Data were not available.
[a]Characteristics are presented as mean ± standard deviation, if not stated otherwise.

stroke, the hazards ratio (HR) for cardiovascular disease compared to the reference (very low risk) group was 1.57 (95% CI = 0.96–2.57, P = 0.069) for the low-risk group, 2.79 (95% CI = 1.70–4.57, P < 0.001) for the moderate-risk group, and 3.87 (95% CI = 2.45–6.11, P < 0.001) for the high-risk group (Table 2).

**Risk predictions in stable and acute chest pain patients.** Furthermore, we tested our deep learning system in outpatients with stable chest pain enrolled and randomized to ECG-gated cardiac CT in the Prospective Multicenter Imaging Study for Evaluation of Chest Pain (PROMISE). In 4021 patients acquired at 193

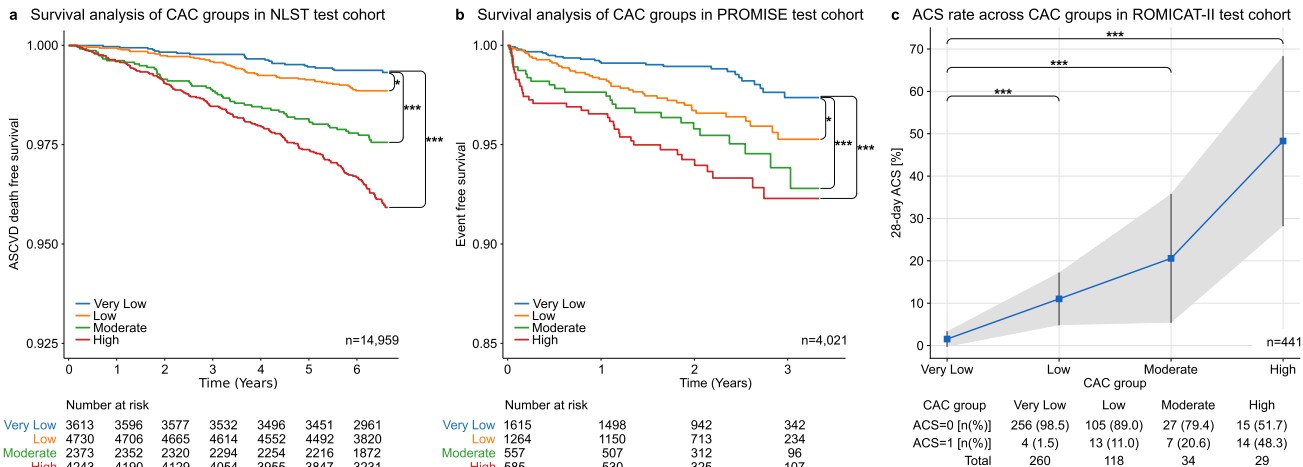

**Fig. 2 Outcome analysis for deep learning predicted calcium scores.** Kaplan–Meier survival analysis of CAC risk groups for **a** cardiovascular disease-related death for 14,959 subjects of the National Lung Screening Trial (NLST)[18] and **b** all-cause mortality, myocardial infarction and unstable angina for 4021 subjects of the Prospective Multicenter Imaging Study for Evaluation of Chest Pain (PROMISE)[19]. A two-sided log rank test was used to calculate the *p* values (*p value ≤ 0.05; ***p value ≤ 0.001) in panels (**a**) and (**b**). **c** Thirty-day acute coronary syndrome (ACS) rate across CAC groups for 441 subjects from the Rule Out Myocardial Infarction using Computer Assisted Tomography II (ROMICAT-II)[20] trial. The shaded area corresponds to the 95% confidence interval of the 30-day ACS rate across CAC groups. A two-sided Fisher's exact test was used to estimate differences in the ACS rate between the very low risk group and the low, moderate, and high risk groups (***p value ≤ 0.001). CAC coronary artery calcium, ASCVD atherosclerotic cardiovascular disease, ACS acute coronary syndrome. CAC risk groups: Very low: 0; Low: 1–100; Moderate: 101–300; High: >300[21].

North American sites, there was a graded association between extent of deep learning calcium score and cardiac events, defined as the composite of death, myocardial infarction, or hospitalization for unstable angina over median 25 months (*P* < 0.001) (Fig. 2b)[19]. After adjustment for Framingham Risk Score (FRS), HRs for cardiac events showed significant increases in hazard across the low, moderate, and high risk versus the reference (very low risk) group (Table 2).

The last test cohort included patients presenting with acute chest pain to the emergency department enrolled in the Rule Out Myocardial Infarction Using Computer Assisted Tomography II (ROMICAT-II) trial. In 441 patients who had ECG-gated cardiac CT at nine sites, there was a similar association between the deep learning calcium score and acute coronary syndrome at 28 days (Fig. 2c)[20]. After adjustment for thrombolysis in myocardial infarction (TIMI) risk score, again patients with increasing deep learning calcium score were at increased risk, reflected in HRs significantly greater than 1 for each of low, moderate, high risk versus the reference (very low risk) group (Table 2). HRs increase as the risk category increases.

**Comparison of automated deep learning and manual results**. We compared the deep learning calcium scores to manually measured calcium scores in 5521 test cohort patients from FHS-CT2 (*n* = 663), NLST (*n* = 396), PROMISE (*n* = 4,021) and ROMICAT-II (*n* = 441). There was a very high[22] Spearman's correlation of 0.92 (*P* < 0.0001) and substantial agreement[23] between automatically and manually calculated calcium risk groups (Fig. 3a). Most differences occurred between adjacent risk categories. For a detailed comparison of the calcium scores in each test cohort, as well as concordance tables and kappa values, see Supplementary Figures 3, 4, and 5 and Supplementary Tables 1 and 3. Furthermore, an in-depth outlier analysis was performed and can be found in the Supplementary Note 1.

To show the predictive value of the automatically calculated calcium score we computed the AUCs for event prediction in NLST, PROMISE, and ROMICAT-II (Supplementary Table 2) and compared them to AUCs from manually derived calcium scores (Supplementary Table 5). We used random effects meta-analysis to estimate combined predicted and manual AUCs. The combined predicted AUC = 0.74 was statistically not different to the combined manual AUC = 0.75 (*P* = 0.544).

**Test–retest reliability**. A test–retest analysis was performed separately on the manual and on the deep learning risk scores on a subset of randomly selected 252 image pairs from FHS-CT1. Each image pair was taken consecutively within the same setup and within 1-min to 1-h time difference. The results showed a great stability between the automatically calculated calcium scores for each image per pair achieving an intra-class correlation (ICC) of 0.993 (*P* < 0.001), compared to the ICC of manual calculated calcium scores of 0.997 (*P* < 0.001). Manual and automatic test–retest repeatability is shown in Fig. 3b, c.

## Discussion

In this investigation, we demonstrate that a deep learning-based coronary calcium scoring system accurately stratifies the risk for cardiovascular events across 19,421 individuals with distinct presentations enrolled in four large clinical studies. Risk prediction was robust across multiple clinical scenarios, including a primary prevention asymptomatic setting with non-gated chest CT (NLST)[18], as well as dedicated ECG-gated cardiac CT in stable (PROMISE)[19] and acute (ROMICAT-II)[20] chest pain setting. The deep learning calcium score in 5521 participants had high correlation with human expert readers and demonstrated robust test–retest reliability. Persons with a calcium score of zero are at very low risk[24], with increasing risk in the ordinal calcium score tiers identified by the deep learning system[25–27]. Based on the 2018 ACC/AHA guidelines[7], in persons at intermediate risk (defined as ≥7.5% to <19.9% 10-year risk of cardiovascular events based on risk factors), a calcium score of 0 indicates very low risk and unlikely benefit from statin therapy, while a high calcium score (≥100 or ≥75th centile for age/sex) indicates that a statin should be considered[7]. Despite these recommendations, a dedicated coronary calcium scoring CT is not yet covered by Medicare and most US insurance companies, and for this reason, there is a great deal of interest in deriving the calcium score from routine chest CTs, which are far more common[8,28,29].

**Table 2 Univariate and multivariable survival analyses of the predictive value of deep learning risk scores assessed in the test cohorts.**

| Risk groups | Events^a | Univariate | | | Multivariable | | | | | |
| --- | --- | --- | --- | --- | --- | --- | --- | --- | --- | --- |
| | | HR | 95% CI | P value | HR | 95% CI | P value | HR | 95% CI | P value |
| **NLST: n = 14,959; events: ASCVD death, n = 288** | | | | | Adjusted for age and sex | | | Adjusted for age, sex, diabetes, hypertension, past heart-disease, past stroke | | |
| Very low | 0.6% (23/3613) | n/a | Reference | Reference | n/a | Reference | Reference | n/a | Reference | Reference |
| Low | 1.1% (53/4730) | 1.77 | 1.08–2.88 | 0.022 | 1.62 | 0.99–2.65 | 0.054 | 1.57 | 0.96–2.57 | 0.069 |
| Moderate | 2.4% (56/2373) | 3.76 | 2.32–6.12 | <0.001 | 3.05 | 1.87–5.00 | <0.001 | 2.79 | 1.70–4.57 | <0.001 |
| High | 3.7% (156/4243) | 5.98 | 3.86–9.26 | <0.001 | 4.34 | 2.75–6.84 | <0.001 | 3.87 | 2.45–6.11 | <0.001 |
| **PROMISE: n = 4,021; events: all-cause mortality, MI, UA, n = 130** | | | | | Adjusted for age and sex | | | Adjusted for Framingham Risk Score | | |
| Very low | 1.5% (25/1615) | n/a | Reference | Reference | n/a | Reference | Reference | n/a | Reference | Reference |
| Low | 3.2% (41/1264) | 2.16 | 1.31–3.55 | 0.002 | 1.96 | 1.18–3.25 | 0.009 | 1.90 | 1.15–3.15 | 0.012 |
| Moderate | 5.0% (28/557) | 3.35 | 1.95–5.74 | <0.001 | 2.83 | 1.62–4.96 | <0.001 | 2.57 | 1.47–4.50 | 0.001 |
| High | 6.2% (36/585) | 4.10 | 2.46–6.84 | <0.001 | 3.21 | 1.84–5.60 | <0.001 | 2.95 | 1.71–5.08 | <0.001 |
| **ROMICAT-II: n = 441; events: ACS, n = 38** | Events | OR | 95%CI | P value | OR | 95%CI | P value | OR | 95% CI | P value |
| | | | | | Adjusted for age and sex | | | Adjusted for TIMI Risk Score | | |
| Very low | 1.5% (4/260) | n/a | Reference | Reference | n/a | Reference | Reference | n/a | Reference | Reference |
| Low | 11.0% (13/118) | 7.92 | 2.52–24.86 | <0.001 | 6.46 | 1.96–21.30 | 0.002 | 7.70 | 2.44–24.30 | 0.001 |
| Moderate | 20.6% (7/34) | 16.59 | 4.56–60.33 | <0.001 | 12.60 | 3.19–49.83 | <0.001 | 16.20 | 4.44–59.13 | <0.001 |
| High | 48.3% (14/29) | 59.73 | 17.50–203.78 | <0.001 | 47.65 | 12.55–180.94 | <0.001 | 57.11 | 16.55–197.12 | <0.001 |

NLST[18] National Lung Screening Trial, PROMISE[19] Prospective Multicenter Imaging Study for Evaluation of Chest Pain, ROMICAT-II[20] Rule Out Myocardial Infarction using Computer Assisted Tomography II, ASCVD atherosclerotic cardiovascular disease, htn hypertension, MI myocardial infarction, UA unstable angina, ACS acute coronary syndrome, OR odds ratio, TIMI thrombolysis in myocardial infarction, n/a not available, Framingham Risk Score: age, total cholesterol, smoker, HDL cholesterol systolic blood pressure. TIMI risk score: age, aspirin use, angina, elevated serum cardiac biomarkers, known coronary artery disease, at least three risk factors for coronary artery disease, such as: hypertension, smoker, low HDL cholesterol, diabetes mellitus, family history of premature coronary artery disease. Coronary calcium risk categories are based on: very low risk (0: no coronary calcifications found), low risk (1–100: small amounts of coronary calcifications), moderate risk (101–300: moderate amounts of coronary calcifications), and high risk (>300: large amounts of coronary calcifications)[2].
^aEvents are presented as a percentage within categorical risk group and with number of events and total number of subjects within the group in parentheses.

Traditionally, coronary calcium scoring requires special software, manual measurement by trained experts and dedicated ECG-gated cardiac CT. As a consequence, the calcium score is often not reported on routine noncardiac chest CT, despite the fact that calcium scores on non-gated CT have reasonably good agreement with dedicated calcium scoring CT[30,31]. Our automated calcium scoring system addresses this need by reliably and accurately extracting the calcium score in both cardiac CT and chest CT. The system calculates the calcium score in under 2 s, without human input. Our approach has several innovations: first, we developed a unique deep learning system to measure CAC on routine cardiac ECG-gated and non-gated chest CT, spanning a broad clinical spectrum including acute and stable chest pain as well as asymptomatic individuals having lung cancer screening. Our analysis of individuals from well-known NIH-sponsored observational cohorts and randomized controlled trials with prospective follow-up for cardiovascular events and death is the largest to date to demonstrate the clinical value of automated calcium scoring. Second, we demonstrate the prognostic value for risk of cardiovascular disease when the deep learning calcium score is applied to four different trials and longitudinal cohorts spanning the range of clinical scenarios in which coronary calcium would be useful. As our deep learning system does not require human input, this makes it an 'end-to-end' solution for accurate and time-efficient cardiovascular risk assessment in clinical settings[32,33]. Third, we share our rigorously validated deep learning system to the public, allowing for accelerated adoption of these technologies by both academic and commercial entities.

Although other studies have investigated deep learning algorithms for automated coronary calcium quantification[34–42], they used smaller cohorts or proprietary technologies. For example, previous publications for fully automatic coronary calcium assessment proposed models focused on either ECG-gated cardiac[34] or non-gated chest CT[35]. Shadmi et al.[40] trained slice based U-Net and FC-DenseNet networks to segment coronary calcium with high accuracy in a subset of NLST, optimizing their model for non-gated CT only. Lessmann et al. presented a two-stage approach for calcium scoring[37] as well as a deep learning method[35] in a smaller subset of NLST. Martin et al.[42] tested in their study a prototype commercial deep learning system for coronary calcium segmentation on a small data set from a single institution and scanner and their median computing time per scan was slightly slower (2.7 s). A combined solution capable of analyzing cardiac and non-gated chest CT as presented in our investigation has only been proposed by de Vos et al.[36] and van Velzen et al.[41]. The approach proposed by de Vos et al.[36] predicted the calcium score directly using direct regression on 2D CT slices only, and their test cohort was substantially smaller compared to our present analysis, less diverse and from the same sites as their training cohort[36]. Van Velzen et al.[41] have shown the automation of CAC measurements compared to manual assessments in several clinical scenarios, using a two-step approach to find calcification candidates and subsequently detecting calcifications, again using smaller testing cohorts compared to our present study. Our analysis of 20,084 individuals from well-known observational cohorts and randomized controlled trials with prospective follow-up for cardiovascular events and death is by far the largest to date to demonstrate the predictive value of automated calcium scoring. Furthermore, we demonstrate strong robustness of the system by a high correlation with manual scoring in 5521 subjects and high test–retest reliability in data from 252 individuals. We also share our deep learning system, including the trained models, with the community, without restrictions.

To overcome different fields of view of CT scans in tested cohorts and to reduce the amount of data that has to be processed to assess coronary calcium, many approaches implement a

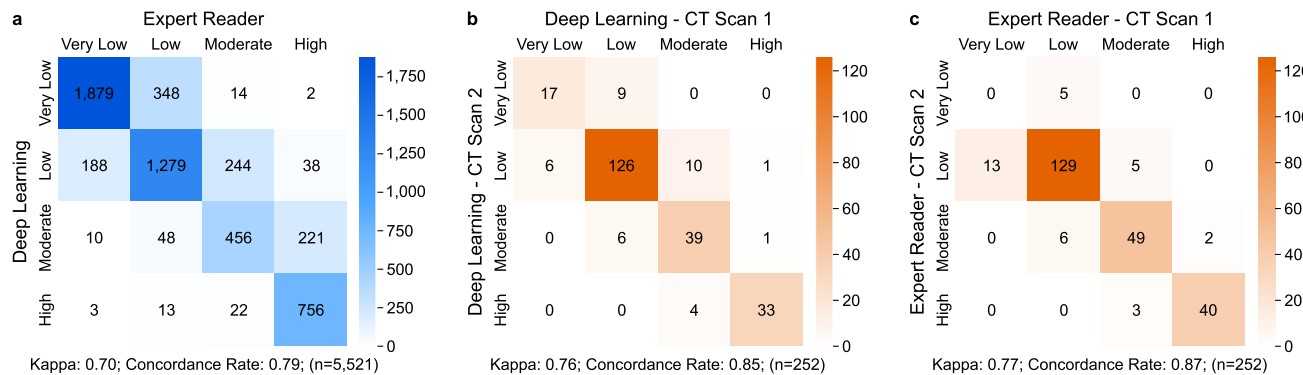

**Fig. 3 Confusion matrices to compare automatic and manual CAC quantification and to assess test–retest repeatability. a** Comparison of CAC classes calculated by the deep-learning framework and expert readers, combining data from FHS-CT2, NLST, PROMISE, and ROMICAT-II (n = 5521). The robustness of **b** the deep learning framework and **c** expert readers to quantify CAC was assessed in 252 FHS-CT1 subjects who underwent two subsequent CT scans within 1 h (Scan 1 and Scan 2). CAC coronary artery calcium, FHS-CT1[17], FHS-CT2[17] Framingham Heart Study, (CT1) participants from the seventh examination cycle of the offspring cohort or first examination cycle of the Third Generation Cohort (2002–05) and (CT2) participants from the second examination cycle of the Third Generation Cohort (2008–11), NLST[18] National Lung Screening Trial, PROMISE[19] Prospective Multicenter Imaging Study for Evaluation of Chest Pain, ROMICAT-II[20] Rule Out Myocardial Infarction using Computer Assisted Tomography II, CAC risk groups: very low: 0; low: 1–100; moderate: 101–300; high: >300[21].

preprocessing step to find a region of interest (ROI) around the heart. Often, traditional image-processing techniques are used to find the ROI[38–40], but also 2D deep learning networks were successfully used to segment the heart and estimate a bounding box[43]. The benefit of our 3D heart segmentation step is not to find a rectangular ROI, but to narrow the region for coronary calcium segmentation to the heart itself.

A strength of our investigation was that we tested our system in populations from large clinical trials and longitudinal cohorts with well-adjudicated cardiovascular disease events. This is essential, as before clinical introduction can be considered, the generalizability of these automated systems needs to be demonstrated as they need to be able to predict cardiovascular events of asymptomatic individuals across multiple clinical scenarios and work robustly on data from multiple institutions. Overall, we included over 20,000 persons drawn from over 200 sites. The available health outcomes and risk factors varied between datasets, reflecting the diverse mix of asymptomatic and symptomatic individuals. Nevertheless, the deep learning calcium score was an independent predictor of adverse cardiovascular events in all cohorts. The majority of FHS (100%)[44], NLST (91%)[18], PROMISE (77%)[45], and ROMICAT-II (66%)[46] participants were non-Hispanic whites. Although the manual calcium score has proven to be an important predictor of cardiovascular events across race and ethnicities, generalizability to other demographics will need to be investigated in future studies[47]. Furthermore, the proposed system evaluated the CAC score on non-contrast cardiac and chest CT. As such it could not detect noncalcified plaque, which can be present even with a calcium score of zero.

In summary, our end-to-end deep learning system provides an automated quantification of coronary calcium on both cardiac CT and lung cancer screening CT. The deep learning calcium score is strongly associated with cardiovascular risk in a broad spectrum of clinical scenarios. Automated quantification of coronary calcium has the potential to improve clinical routine and population health.

## Methods

**Study population.** This study was a retrospective secondary analysis of a longitudinal primary prevention cohort (FHS-CT1 and FHS-CT2) and three randomized clinical trials (NLST, PROMISE, ROMICAT–II). Details about participant selection are provided in the consort diagrams in the Supplementary Fig. 1.

The deep learning system training and tuning was accomplished in FHS Offspring[48] and Third Generation[49] cohort participants (FHS-CT1, n = 1636) who had non-contrast ECG-gated cardiac CT for coronary calcium quantification between 2002 and 2005. Details regarding the FHS cohort, inclusion criteria, and calcium scoring have been described elsewhere[21]. Participants resided or had parents who resided in Framingham or in the New England region were drawn for this study. Major inclusion criteria were age ≥35 years for men and ≥40 years for women. All participants provided written consent for the CT study, which was approved by the institutional review boards of the Boston University Medical Center and Massachusetts General Hospital[17,21]. In our investigation we included only participants with available cardiovascular disease risk profile, no known prior cardiovascular disease, and diagnostic-quality cardiac CT as determined by an expert reader (Supplementary Fig. 1a).

The deep learning system performance was tested in a second, independent group of FHS participants who had cardiac CT from 2008 to 2011 (FHS-CT2, n = 663). None of the persons in the FHS-CT2 testing cohort were in the FHS-CT1 training/tuning cohort. While the FHS-CT1 training cohort included only diagnostic-quality CTs, the FHS-CT2 testing cohort included all CTs including those initially considered non-diagnostic (Supplementary Fig. 1b).

A second testing cohort was drawn from the NLST[18], a multicenter randomized controlled trial of non-contrast, non-ECG-gated low-dose chest CT for lung cancer screening. In NLST, 53,454 subjects aged 55–74 years, current or former heavy smokers, were enrolled at 33 participating medical institutions with all-cause mortality as the primary outcome measure over a follow-up of up to 8 years. A total of 26,722 randomly selected participants underwent low-dose non-contrast chest CT imaging between 2002 and 2007. The trial was approved by the institutional review board at each site. From the full cohort we had permission to include 15,000 randomly selected subjects. For each subject the baseline (T0) CT scan was chosen with soft kernel preferred over hard kernel reconstructed images. We excluded 17 subjects that did not have a T0 scan, 10 subjects that did not have scans which met our quality requirements, 12 subjects that had a broken or incomplete scan, and 2 subjects with missing risk data. The final testing cohort consisted of 14,959 scans (Supplementary Fig. 1c). To verify the results of our deep learning system in this cohort, a subset of randomly chosen 396 subjects were segmented by expert readers.

The third cohort included participants from the Prospective Multicenter Imaging Study for Evaluation of Chest Pain (PROMISE)[19,45]. In this multicenter trial 10,003 symptomatic patients were randomized at 193 medical sites in North America using a composite of major cardiovascular events as a primary outcome measure over a median follow-up of 25 months. Participants of age 45 to 64 years with stable chest pain and without known prior CAD were enrolled between 2010 and 2013 with 4,996 subjects randomly selected to undergo cardiac CT imaging. The central activities of the study were approved by the Duke, Partners Healthcare, and Tufts Institutional Review Boards. Furthermore, local or central IRBS approved the study at each medical institution. The final testing cohort included 4,021 individuals each with a non-contrast cardiac CT scan and full risk profile available (Supplementary Fig. 1d). All subjects were segmented by expert readers.

The fourth cohort included participants from the Rule Out Myocardial Infarction Using Computer Assisted Tomography Study Two (ROMICAT-II)[20]. In this randomized open-labeled multicenter trial 1000 patients which presented at the emergency department of nine clinical sites with acute chest pain were enrolled between 2010 and 2012. The primary outcome measure of this study was the length

of the hospital stay and a second outcome including undetected acute coronary syndrome within 72 h after hospital discharge, increased adverse events, major adverse cardiovascular events within 28 days, and periprocedural complications. The study was approved by the local institutional review boards. Patients were between 40 and 74 years old, without known coronary artery disease, almost equal gender representation and significant representation of all minorities. Of these subjects, 500 were randomly selected to undergo non-contrast cardiac CT imaging. After excluding participants with incomplete image data or risk profile the final testing cohort included 441 participants (Supplementary Fig. 1e). All subjects were segmented by expert readers.

A detailed population description for all four cohorts can be found in Table 1. Participants from all studies provided written consent.

**Deep learning-based coronary calcium segmentation**. We propose a deep learning system that is able to automatically calculate a calcium score from a given CT scan for cardiovascular risk prediction. The system consists of four consecutive steps for (1) heart localization, (2) heart segmentation, (3) coronary calcium segmentation, and (4) calcium score calculation. We trained a separate fully convolutional neural network of the U-Net[50] architecture for each of the first three steps. The U-Net architecture was originally designed for biomedical image segmentation with the goal of overcoming the requirement for a very large cohort for training a deep learning network.

The cohort for training and tuning the three deep learning models consisted of 1636 CT scans: 623 CT scans were from subjects with coronary calcium and 1013 CT scans from subjects with no coronary calcium. Although several hundred more CT scans from subjects with no coronary calcium were available, we chose to exclude them to keep the imbalance between subjects with and without coronary calcium small. Excluded subjects were selected randomly. Coronary calcium, if present, was manually segmented by experienced readers in all subjects. Furthermore, the heart was manually segmented in a subset of 129 randomly selected subjects of the training cohort. Our testing cohort consisted of 20,084 subjects from four different clinical studies and trials including dedicated cardiac CT scans as well as lung screening CT scans, health outcomes, and follow-up information. Manually calculated calcium scores from expert readers were available for 5521 subjects and manually segmented hearts for 895 subjects. All CT scans were padded and cropped to the same size of $512 \times 512 \times 512$ pixel (px) and resampled to the same resolution of $0.7 \times 0.7 \times 2.5$ mm/px. A detailed description of the training, tuning and testing cohorts and their usage is described in the Supplementary Fig. 2a.

The first network in our system was trained to localize the heart within a given 3D CT scan. This step was necessary as CT scans could differ, for example, in size, resolution, area captured, or field of view, depending on the cohort, scanner used, and site acquiring the scan. The training cohort was split 70/30% for training and tuning, and all scans were downsampled to a size of $112 \times 112 \times 112$ px to fit into the GPU memory. The model used for training was a standard U-Net with four downsampling steps running for 1200 epochs. Data augmentation was used by applying rotation of ±4 degrees around the sagittal, transversal, and longitudinal axis for heart localization and ±35 degrees around the sagittal axis for heart segmentation. Furthermore, we applied translation within ±10px in the axial plane for heart localization and ±20px in the axial plane for heart segmentation. The output of the network was upsampled to the initial CT scan size, leading to a very rough heart segmentation that we used for placing a bounding box for the subsequent steps.

The second network of the deep learning system was trained to segment the heart. The input scans were first cropped to $384 \times 384 \times 80$ px cubes around the heart center and then downsampled to $128 \times 128 \times 80$ px. The training cohort was again split 70/30% for training and tuning and data was augmented by applying rotations and translations in small ranges only. The model used for training had the same architecture as in the previous step with four downsampling steps running for 1000 epochs. Once the model parameters were found to perform well on the tuning cohort, the final model was trained combining the training and tuning cohorts for better performance. The output of the network was upsampled to the initial CT scan size, leading to an accurate heart segmentation. As this step was mainly to reduce the area for the consecutive calcium segmentation step and although the error of the heart segmentation was low, we added a rim of 11 pixels to the predicted heart segmentation to ensure the whole heart was captured.

The third network was trained to segment coronary calcium. For this step, we divided the previously segmented heart into smaller cubes of size $48 \times 48 \times 32$ px. Extensive testing of several cube sizes showed the chosen size worked best as larger cubes increased the training time while the accuracy stayed the same. We used cubes overlapping all but one pixel for the training, whereas cubes did not overlap during the testing. The model used in this step was a U-Net[50] with three downsampling steps extended by batch normalization layers in the contracting path (left side) for better generalizability (Supplementary Fig. 2b). The resulting segmentation patches were aligned again, leading to a coronary calcium segmentation of the heart. The final step was to threshold the whole segmentation by 0.95 to obtain the binary calcium mask.

With the coronary calcium segmented, the calcium score was calculated using a volumetric implementation of the method by Agatston and Janowitz[5]. The calcium score was calculated by multiplying the volume of a coronary calcification with a

factor, depending on the highest density within the calcification, with the density being measured in Hounsfield Units (HU). This weight factor was 1 for a density between 130 and 199 HU, 2 for 200 to 299 HU, 3 for 300 to 399 HU, and 4 for 400 HU and above. Calcification with a volume below one cubic-millimeter was considered noise and excluded from the calculation. The final calcium score per patient was the sum of the weighted calcifications. For further analysis we stratified the calcium risk score into the risk groups very low (0), low (1–100), moderate (101–300), and high (>300)[21].

Training, tuning, and testing was done on a Linux workstation using Tensorflow-GPU and Keras. The only notable hardware requirement was to have 4 GPUs with at least 64 gigabytes of memory to fit a reasonable batch size of input volumes for the heart segmentations.

**Technical evaluation**. The performance of the deep learning system was tested by reviewing CT scans with discordances between the manual and deep learning coronary calcium categories (Supplementary Fig. 7). Most discrepancies were due to misclassification of non-coronary calcium as coronary calcium and vice versa. In a few instances, inaccurate heart segmentation led to coronary calcium being outside the ROI of the calcium segmentation network and hence being missed. Furthermore, we measured the time the system needed to process scans. On average, the deep learning system assessed the CAC score in under two seconds per scan.

**Manual calcium score assessment**. The CAC score was measured manually by expert readers using the method by Agatston and Janowitz[5,6] on dedicated workstations, as reported in the parent FHS[17], PROMISE[19], and ROMICAT-II[20] studies. In NLST[18], the coronary calcium score was measured manually in 396 randomly selected participants using 3D Slicer (V4).

**Test–Retest analysis**. In FHS-CT1, 252 participants underwent cardiac ECG-gated CT twice within 1 h to assess test–retest reliability. The deep learning system and the human readers quantified calcium on both scans to assess test–retest reliability.

**Statistical methods**. In this study we described continuous variables as the mean ± standard deviation (SD) and categorical variables as frequencies and percentages. Furthermore, we performed univariate and multivariate Cox proportional hazards regressions comparing cardiovascular disease risk in the 1st (lowest) versus the 2nd–4th quartiles of CAC. The dependent variable in FHS-CT2 was a composite of cardiovascular disease event and all-cause mortality with a mean follow-up of 8.8 years[51]. Events in NLST were defined as ASCVD mortality with a follow-up of up to 9 years[18]. Events in PROMISE were defined as a composite of all-cause mortality, myocardial infarction, major complications from cardiovascular procedures and diagnostic testing and unstable angina with an average follow-up of 2.5 years[19]. Events in the ROMICAT-II trial were defined as major adverse cardiovascular events in a time frame of 28 days after admission to the emergency room[46]. Cox proportional hazards models and log rank tests were used to estimate and compare hazard ratios between the reference (very low risk) group (0: no coronary calcifications found), the low risk group (1–100: small amounts of coronary calcifications), the moderate risk group (101–300: moderate amounts of coronary calcifications), and the high risk group (>300: large amounts of coronary calcifications): one unadjusted model, one model adjusted for age and sex, and a third model in which we additionally adjusted for standard cardiovascular risk factors (NLST: hypertension, diabetes, past coronary artery disease, past stroke; PROMISE: FRS; ROMICAT-II: TIMI risk score) using Survival R package (v3.2-3). Standard Kaplan–Meier survival curves were generated to visualize event-free survival for the NLST and PROMISE testing cohorts in R using the Survminer package (v0.4.8). The log rank test was used to identify significant differences in survival. To assess the similarity of automatically and manually derived calcium scores, we calculated the Spearman's correlation (Python package scipy.stats. spearmanr based on Zwillinger, D. and Kokoska[52]), the ICC, which was calculated from components of a one-way analysis of variance in R using the ICC package (v2.3.0) based on Searle[53], Donner[54] and Thomas and Hultquist[55], the Cohen's Kappa[56] (Python package sklearn.metrics.cohen_kappa_score) and the concordance rate (calculated as number of concordant pairs divided by the number of all pairs). The combined AUCs were estimated and compared using the survcomp R package (V1.36.1). BMI was calculated using the weight (pounds) and height (inches) as weight/height$^2$ * 703[57].

**Reporting summary**. Further information on research design is available in the Nature Research Reporting Summary linked to this article.

## Data availability

NLST data including raw CT images may be requested from the National Cancer Institute (https://biometry.nci.nih.gov/cdas/nlst/). Although raw CT imaging data cannot be shared, all measured results to replicate the statistical analysis in NLST are shared at the AIM webpage at aim.hms.harvard.edu/deepcac. Furthermore, we include test samples from a publicly available data set with deep learning and expert reader heart and calcium segmentations.

## Code availability

The code of the deep learning system, as well as the trained model and statistical analysis are publicly available at the AIM webpage aim.hms.harvard.edu/deepcac.

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

## Acknowledgements

The authors thank the Framingham Heart Study, NCI, ACRIN, NLST, Prospective Multicenter Imaging Study for Evaluation of Chest Pain, and Rule Out Myocardial Infarction Using Computer Assisted Tomography II trial for access to trial data. The authors acknowledge financial support from NIH (HA: NIH-USA U24CA194354, NIH-USA U01CA190234, NIH-USA U01CA209414, and NIH-USA R35CA22052; UH: NIH, 5R01-HL109711, NIH/NHLBI 5K24HL113128, NIH/NHLBI 5T32HL076136, NIH/NHLBI 5U01HL123339), the European Union—European Research Council (HA: 866504), as well as the German Research Foundation (DFG; TA: 1438/1-1 and WE: 6405/2-1), American Heart Association Institute for Precision Cardiovascular Medicine (MTL: 18UNPG34030172), Fulbright Visiting Researcher Grant (E0583118), Rosztoczy Foundation Grant. The Framingham Heart Study (FHS) acknowledges the support of contracts NO1-HC-25195, HHSN268201500001I, and 75N92019D00031 from the National Heart, Lung and Blood Institute.

## Author contributions

Study design: R.Z., B.F., P.E., J.W., I.A., J.T., C.P., U.H., M.T.L., H.J.W.L.A.; code design, implementation and execution: R.Z.; acquisition, analysis or interpretation of data: R.Z., B.F., P.E.; image segmentation: B.F., P.E., J.W., I.A., J.T., R.M.A., D.B., M.U., Y.K., J.K., L.Z., J.E.S.; writing of the manuscript: R.Z., B.F., J.W., U.H., M.T.L., H.J.W.L.A.; critical revision of the manuscript for important intellectual content: all authors; statistical analysis: R.Z., B.F., M.T.L., A.L., J.M.M., T.M.; study supervision: U.H., M.T.L., H.J.W.L.A.

## Competing interests

The authors declare no competing interests.
