## [Peer Review File · Nature Communications]

Reviewers' Comments:

Reviewer #1:

Remarks to the Author:

The first contribution study is to validate the performance of deep learning on coronary calcium segmentation and scoring, by more than 5,000 subjects with manual CAC annotation. The second contribution is to associate the CAC scores with clinical outcomes using by far the largest 20,000 subjects. The major claims are (1) the deep learning scores are high correlation with manual quantification, with robust test-retest reliability. (2) the deep learning outcome is a strong predictor of cardiovascular events as risk factors.

The advantages of this study are obvious. First, the sample size is by far the largest. Second, the paper is well-written with comprehensive statistical analysis. Third, the method is validated on different cohorts including NLST, chest pain cohort etc. Four, it provides a comprehensive baseline that proves deep learning based CAC scoring is clinically relevant for risk prediction.

The biggest concern is the lack of novelty, in terms of clinical conclusions, and methodology. For clinical conclusion, there has been quite a lot of previous clinical studies that have investigated the effectiveness of CAC scores for different cohorts. Also, many recent technical studies have investigated the effectiveness of deep learning for CAC scoring. So, from previous studies, the conclusions have already been made that (1) deep learning is robust for CAC segmentation and scoring, (2) CAC scores are a strong predictor for CAD events. Therefore, limited new conclusion or finding is provided. For methodology, the three U-Net models were trained to perform coarse to fine CAC segmentation. Yet, the U-Net and the training strategy are not new.

Following are some specific comments:

From the validation using 5,521 subjects with manual segmentation, we can know how good the deep learning segmentation is compared with human segmentation. But, we don't know what the clinical difference is between using deep learning and using manual results. It would be interesting to see the difference of clinical prediction and prognosis results, between using 5,521 manual CAC scores and using 5,521 auto CAC scores. Then that would give us a better sense for clinical relevance than segmentation performance.

In supplementary FS3, it is not surprising to see lots of extreme cases that the expert predicts zero while the deep learning predicts non-zero, while the expert predicts non-zero while the deep learning predicts zeros. Some cases have been shown in FS7, however, more comprehensive analysis and explanations on those cases will be very valuable for other researchers to understand the limitation/advantage of deep learning. That would be very interesting as the cohorts in this study are large and heterogeneous.

In Table S2, the AUC is provided. The AUC is typically for binary classification, so it is not quite clear where the AUC is calculated from.

The third U-Net was trained to segment coronary calcium. The author divided the previously segmented heart 384x384x80px into smaller cubes of size 48x48x32 pixels with a sliding window design. But this might not be the best practice. In CAC segmentation, the most challenging part is to distinguish the CAC from other calcium or artifacts with HU>300. For that purpose, we typically would like to see the global arteries and see if CAC are in arteries. However, the small cube would lose such global spatial information for distinguishing CAC, that would also be the reason that the Dice and Pearson correlation are not quite high. Probably a 2.5D design, with full heart region in plane and a couple of slices on z direction, might be a better solution.

Reviewer #2:

Remarks to the Author:

The authors present a study entitled "Deep convolutional neural networks to predict cardiovascular risk from computed tomography". In this study, the authors evaluated a deep-learning system to quantify coronary calcium on gated and non-gated cardiac CT scans. They could demonstrate feasibility, high correlation with manual measurements, robust test-retest reliability and a strong prediction of cardiovascular events, when applied to three different cohort studies.

Overall, this is a well written manuscript. The topic is scientifically interesting and relevant, nevertheless there are major weaknesses of this work (see below):

Major Strengths:

- Training of the prediction system in an epidemiological cohort, validation of the prediction system in three (mostly) large-scale cohorts, covering gated and non-gated CT scans, a broad range of disease likelihood of coronary calcifications, and scanners / protocols from different vendors / centers.
- Outcome data
- Testing vs manual segmentations and test-re-test data

Major Weaknesses:

Novelty:

- A) Several other groups have already published the feasibility of using deep learning / convolutional neural networks (CNN) to quantify coronary calcium using cardiac gated and /or non-gated CTs, e.g.:

van Velzen SGM, Lessmann N, Velthuis BK, Bank IEM, van den Bongard D, Leiner T, et al. Deep Learning for Automatic Calcium Scoring in CT: Validation Using Multiple Cardiac CT and Chest CT Protocols. *Radiology*. 2020;191621. doi: 10.1148/radiol.2020191621.

de Vos BD, Wolterink JM, Leiner T, de Jong PA, Lessmann N, Işgum I. Direct Automatic Coronary Calcium Scoring in Cardiac and Chest CT. *IEEE transactions on medical imaging*. 2019;38(9):2127-38. doi: 10.1109/tmi.2019.2899534.

Lessmann N, Işgum I, Setio AA, de Vos B, Ciompi F, de Jong P, et al. Deep convolutional neural networks for automatic coronary calcium scoring in a screening study with low-dose chest CT. *SPIE Medical Imaging*. SPIE; 2016.

Martin SS, van Assen M, Rapaka S, Hudson HT, Jr., Fischer AM, Varga-Szemes A, et al. Evaluation of a Deep Learning-Based Automated CT Coronary Artery Calcium Scoring Algorithm. *JACC Cardiovascular imaging*. 2020;13(2 Pt 1):524-6. doi: 10.1016/j.jcmg.2019.09.015.

Lessmann N, van Ginneken B, Zreik M, de Jong PA, de Vos BD, Viergever MA, et al. Automatic Calcium Scoring in Low-Dose Chest CT Using Deep Neural Networks With Dilated Convolutions. *IEEE transactions on medical imaging*. 2018;37(2):615-25. doi: 10.1109/tmi.2017.2769839.

- In addition, other previous publications have used their CNN in dedicated populations with or without cardiac gating. The here presented approach is a combination that works on both, as also published by van Velzen et al. (this work is not cited or discussed by the authors) . Despite the application to large cohort studies, there is a lack of novelty in this approach. However, I agree with the authors that their work is overall a step in the right direction.

- B) The prognostic implications of coronary calcium quantification on prognosis are well known, especially for most of the used cohorts in this study. The quantification by using CNN could definitively support clinical value, but the results to predict outcome is already well documented by

the authors and others , as cited in the manuscript.

Minor:

- C) The scoring system is demonstrated in US populations, validation in European or Asian cohorts is not existing.
- D) Comparable to manual calcium scoring, this approach will miss all patients without calcified plaques, e.g. patients with vulnerable soft plaques. Quantitative analysis of plaques and stenosis (not only calcifications) by CNN would address this need.
- E) The authors are not demonstrating equality / superiority of outcome prediction compared to manual measurements.

Point-by-point response to the questions raised by the reviewers

Reviewer #1 (Remarks to the Author):

The first contribution study is to validate the performance of deep learning on coronary calcium segmentation and scoring, by more than 5,000 subjects with manual CAC annotation. The second contribution is to associate the CAC scores with clinical outcomes using by far the largest 20,000 subjects. The major claims are (1) the deep learning scores are high correlation with manual quantification, with robust test-retest reliability. (2) the deep learning outcome is a strong predictor of cardiovascular events as risk factors.

The advantages of this study are obvious. First, the sample size is by far the largest. Second, the paper is well-written with comprehensive statistical analysis. Third, the method is validated on different cohorts including NLST, chest pain cohort etc. Four, it provides a comprehensive baseline that proves deep learning based CAC scoring is clinical relevant for risk prediction.

We would like to thank the referee for these very positive remarks.

The biggest concern is the lack of novelty, in terms of clinical conclusions, and methodology. For clinical conclusion, there has been quite a lot of previous clinical studies have investigated the effectiveness of CAC scores for different cohorts. Also, many recent technical studies have investigated the effectiveness of deep learning for CAC scoring. So, from previous studies, the conclusions have already been made that (1) deep learning is robust for CAC segmentation and scoring, (2) CAC scores are a strong predictor for CAD events. Therefore, limited new conclusion or finding is provided. For methodology. The three U-Net models were trained to perform coarse to fine CAC segmentation. Yet, the U-Net and the training strategy are not new.

Key innovations of our work are:

1. We demonstrate, for the first time, that deep learning can accurately measure coronary calcium and predict cardiovascular events in 14,959 NLST, 4,021 PROMISE, and 441 ROMICAT-II participants, spanning a broad clinical spectrum including acute and stable chest pain as well as asymptomatic persons having lung cancer screening. Our analysis of 20,084 individuals from well-known NIH-sponsored observational cohorts and randomized controlled trials with prospective followup for cardiovascular events and death, is the largest to date to demonstrate the clinical value of automated calcium scoring. Most of these previous studies used the manual CACS measurement as the reference standard; our study takes this an important step further to demonstrate that the automated CACS predicts prospectively acquired cardiovascular events.
2. Although automated systems have been published before, none of them have been validated to predict cardiovascular events or could work robustly in several clinical implementation areas, ranging from cardiac gated CT scans to non-gated lung screening scans.
3. We are the first to open source the code for our automated deep learning model with the open MIT license. Free open source distribution is critical for this field, and will allow other investigators to use the model on their datasets and serve as the benchmark for future research. To our knowledge, all previous automated CACS tools have not been open sourced.

These important points are added to the discussion of the revised manuscript (Page 7; Line 196):

Our approach has several innovations: first, we developed a unique deep learning system to measure coronary artery calcium on routine cardiac electrocardiography (ECG)-gated and non-gated chest CT, spanning a broad clinical spectrum including acute and stable chest pain as well as asymptomatic individuals having lung cancer screening. Our analysis of individuals from well-known NIH-sponsored observational cohorts and randomized controlled trials with prospective followup for cardiovascular events and death, is the largest to date to demonstrate the clinical value of automated calcium scoring. Second, we demonstrate prognostic value for risk of cardiovascular disease when the deep learning calcium score is applied to four different trials and longitudinal cohorts spanning the range of clinical scenarios in which coronary calcium would be useful. As our deep learning system does not require human input, this makes it an 'end-to-end' solution for accurate and time-efficient cardiovascular risk assessment in clinical settings^{32,33}. Third, we share our rigorously validated deep learning system to the public, allowing for accelerated adoption of these technologies by both academic and commercial entities.

Following are some specific comments:

From the validation using 5,521 subjects with manual segmentation, we can know how good the deep learning segmentation is compared with human segmentation. But, we don't know what the clinical difference is between using deep learning and using manual results. It would be interesting to see the difference of clinical prediction and prognosis results, between using 5,521 manual CAC scores and using 5,521 auto CAC scores. Then that would give us a better sense for clinical relevance than segmentation performance.

We agree with the reviewer that such an analysis is valuable. We added a comparison of the prognostic value of automatic versus manual scoring to the revised manuscript. There was no significant difference in AUC for cardiovascular events between manual and automatic CAC scores (see results section *Comparison of automated deep learning to manual coronary artery calcium scoring and event prediction* (Page 6; Line 154), as well as **Table S5**; Supplements).

Comparison of automated deep learning to manual coronary artery calcium scoring and event prediction.

[...]

*To show the predictive value of the automatically calculated calcium score we computed the AUCs for event prediction in NLST, PROMISE and ROMICAT-II (Table S2) and compared them to AUCs from manually derived calcium scores (**Table S5**). We used random effects meta-analysis to estimate combined predicted and manual AUCs. The combined predicted AUC=0.74 was statistically equal to the combined manual AUC=0.75 (P=0.544).*

In supplementary FS3, it is not surprising to see lots of extreme cases that the expert predicts zero while the deep learning predicts non-zero, while the expert predicts non-zero while the deep learning predict zeros. Some cases have been shown in FS7, however, more comprehensive analysis and explanations on those cases will be very valuable for other researchers to understand the limitation/advantage of deep learning. That would be very interesting as the cohorts in this study are large and heterogeneous.

We agree with the reviewer that an in-depth assessment of extreme cases could be interesting for other researchers and future application of our network. We manually examined the 32 cases with the highest AI to manual CAC score difference and included these new results to the **Supplementary Section S1** of the revised manuscript:

Section S1. Extended failure and outlier analysis

To get more insight in the performance of the proposed deep learning system we conducted an in-depth analysis of cases with extreme differences between automatically predicted and manual calcium scores. Therefore we looked into subjects where the predicted calcium score was zero and the human calcium score was above 100 or vice versa, which was the case in 32 out of 5,521 subjects.

In the majority of the cases the deep learning system missed a plaque ($n=17$), where in one case the missed plaque was close to the valve, one case had severe motion artifacts and in the rest of the cases no obvious reason could be found for the misses. In eight cases the deep learning system segmented a wrong calcium object. In three of these cases noise was segmented in scans of poor image quality, in further three cases valve calcification was segmented, in one case a metal artifact was segmented and in one case a lymph node close to the heart was segmented.

In four cases the human reader missed calcium which were segmented correctly by the deep learning system. In one case the human reader segmented a wrong calcium object while the deep learning system correctly did not segment it. In one case we accidentally included a contrast enhanced scan in our test cohort and the deep learning system segmented a wrong area. Unfortunately, this case was not excluded in our preprocessing image quality assessment. One case had very poor image quality with high noise and motion artifacts which led to human over-segmenting and deep learning under-segmenting coronary calcium.

Manual assessment of segmentation results showed the most common area for segmentation errors was near the aortic and mitral valves. Distinguishing coronary and valvular calcium can be challenging even for humans on ECG-gated CTs.

As shown in **Figure S7**, the proposed system was able to handle metal artifacts in CT scans in most but not all cases. Current clinical guidelines recommend applying the coronary artery calcium score in patients without known cardiovascular disease⁷, and the value of the calcium score in patients with metal in the heart from past cardiac interventions (e.g. prior coronary artery bypass graft, coronary artery stents, or pacemakers) is not clear.

In Table S2, the AUC is provided. The AUC is typically for binary classification, so it is not quite clear where the AUC is calculated from.

We used the AUC to classify binary events (cardiovascular event or no event), based on the calculated continuous CAC scores. We have modified the **Table S2** legend to clarify this important point:

Table S2. AUC values for event prediction based on automatically calculated calcium scores.

The third U-Net for was trained to segment coronary calcium. Author divided the previously segmented heart 384x384x80px into smaller cubes of size 48x48x32 pixels with a sliding window design. But this might not be the best practice. In CAC segmentation, the most challenging part is to distinguish the CAC from other calcium or artifacts with $HU>300$. For that purpose, we typically would like to see the global arteries and see if CAC are in arteries. However, the small cube would lose such global spatial information for distinguishing CAC, that would also be the reason that the Dice and Pearson correlation are not quite high. Probably a 2.5D design, with full heart region in plane and a couple of slices on z direction, might be a better solution.

We thank the reviewer for this very interesting question. Here are additional details we experienced during the development of the deep learning system:

- The cube size, although small, turned out to be sufficiently large enough to identify calcium as coronary calcium or not. During our development we trained several models using varying cube sizes. The increased cube size did not lead to a better performance.

- From assessing our results during the development process, bones were rarely falsely segmented, most likely due to their shape difference to coronary calcium.
- Metal artifacts are an interesting problem but as they have usually extremely high HU values, 1000 and higher, we did not encounter many false segmentations. We show examples in figure S7.
- From what we saw in our results, the area where our system had the most problems was at the valve. Especially in non gated scans, it is often hard to exactly see where the valve ends and where coronary arteries begin which is also challenging for human readers.
- Regarding the performance of our 3D model versus a 2.5D model, this is a very interesting topic but we think this should be part of a separate technical study. Our used model showed great performance in CAC classification and event prediction and we are certain that this work presents a baseline for future deep learning models.

We updated the manuscript and added these points to the revised methods section (Page 12; Line 353) and the supplementary materials:

Deep learning based coronary calcium segmentation.

[...]

The third network was trained to segment coronary calcium. For this step, we divided the previously segmented heart into smaller cubes of size 48×48×32px. Extensive testing of several cube sizes showed the chosen size worked best as larger cubes increased training time while the accuracy stayed the same.

Section S1. Extended failure and outlier analysis

[...]

Manual assessment of segmentation results showed the most common area for segmentation errors was near the aortic and mitral valves. Distinguishing coronary and valvular calcium can be challenging even for humans on ECG-gated CTs.

*As shown in **Figure S7**, the proposed system was able to handle metal artifacts in CT scans in most but not all cases. Current clinical guidelines recommend applying the coronary artery calcium score in patients without known cardiovascular disease⁷, and the value of the calcium score in patients with metal in the heart from past cardiac interventions (e.g. prior coronary artery bypass graft, coronary artery stents, or pacemakers) is not clear.*

Reviewer #2 (Remarks to the Author):

The authors present a study entitled “Deep convolutional neural networks to predict cardiovascular risk from computed tomography”. In this study, the authors evaluated a deep-learning system to quantify coronary calcium on gated and non-gated cardiac CT scans. They could demonstrate feasibility, high correlation with manual measurements, robust test-retest reliability and a strong prediction of cardiovascular events, when applied to three different cohort studies.

We would like to thank the referee for these positive remarks.

Overall, this is a well written manuscript. The topic is scientifically interesting and relevant, nevertheless there are major weaknesses of this work (see below):

Major Strengths:

- Training of the prediction system in an epidemiological cohort, validation of the prediction system in three (mostly) large-scale cohorts, covering gated and non-gated CT scans, a broad range of disease likelihood of coronary calcifications, and scanners / protocols from different vendors / centers.
- Outcome data
- Testing vs manual segmentations and test-re-test data

We thank the reviewer for these positive remarks.

Major Weaknesses:

Novelty:

- A) Several other groups have already published the feasibility of using deep learning / convolutional neural networks (CNN) to quantify coronary calcium using cardiac gated and /or non-gated CTs, e.g.:

1. van Velzen SGM, Lessmann N, Velthuis BK, Bank IEM, van den Bongard D, Leiner T, et al. Deep Learning for Automatic Calcium Scoring in CT: Validation Using Multiple Cardiac CT and Chest CT Protocols. *Radiology*. 2020;191621. doi: 10.1148/radiol.2020191621.
2. de Vos BD, Wolterink JM, Leiner T, de Jong PA, Lessmann N, Isgum I. Direct Automatic Coronary Calcium Scoring in Cardiac and Chest CT. *IEEE transactions on medical imaging*. 2019;38(9):2127-38. doi: 10.1109/tmi.2019.2899534.
3. Lessmann N, Işgum I, Setio AA, de Vos B, Ciompi F, de Jong P, et al. Deep convolutional neural networks for automatic coronary calcium scoring in a screening study with low-dose chest CT. *SPIE Medical Imaging*. SPIE; 2016.
4. Martin SS, van Assen M, Rapaka S, Hudson HT, Jr., Fischer AM, Varga-Szemes A, et al. Evaluation of a Deep Learning-Based Automated CT Coronary Artery Calcium Scoring Algorithm. *JACC Cardiovascular imaging*. 2020;13(2 Pt 1):524-6. doi: 10.1016/j.jcmg.2019.09.015.
5. Lessmann N, van Ginneken B, Zreik M, de Jong PA, de Vos BD, Viergever MA, et al. Automatic Calcium Scoring in Low-Dose Chest CT Using Deep Neural Networks With Dilated Convolutions. *IEEE transactions on medical imaging*. 2018;37(2):615-25. doi: 10.1109/tmi.2017.2769839.

We agree with the referee that other ML studies have been published, although in much smaller datasets. While the study of van Velzen et.al includes data from several different CT examination types, the overall number of scans included was much smaller (we included data of 12844 more individuals) and without any outcome analyses compared to our study. The work of de Vos is a technical feasibility study for direct calcium score prediction using deep learning. Their training and test sets, although independent, were subsets of the same cohort and their test set was again much smaller than ours. The study of Lessmann

et.al included 1028 scans from three sites only (We had data acquired at 236 different sites), and their training and test sets were subsets of the same cohort as well. Although Martin et.al had an independent test set, it consisted of 511 cases only which were acquired on one CT scanner. The study of Lessmann et al. was trained and tested on a small subset of NLST data only. None of these studies included clinical outcomes analyses. Therefore there is an important translational gap as no studies have shown that a deep learning system is able to predict cardiovascular events and work robustly in several clinical areas.

We address this gap by demonstrating that a single deep learning system can accurately measure and predict cardiovascular events in large diverse and well-known observational cohorts and randomized controlled trials. Furthermore, we are the first to open source our model. This important step will allow other investigators to apply it to their datasets and serve as the benchmark for future research and accelerate clinical adoption.

We added the two missing studies to the discussion of the revised manuscript (Page 8; Lines 215 and 222):

Martin et al.⁴² tested in their study a prototype commercial deep learning system for coronary calcium segmentation on a small data set from a single institution and scanner and their median computing time per scan was slightly slower (2.7 seconds).

A combined solution capable of analyzing cardiac and non-gated chest CT as presented in our investigation has only been proposed by de Vos et al.³⁶ and van Velzen et al.⁴¹. [...] Van Velzen et al.⁴¹ has shown the automation of CAC measurements compared to manual assessments in several clinical scenarios, using a two step approach to find calcification candidates and subsequently detecting calcifications, again using smaller testing cohorts compared to our present study.

- In addition, other previous publications have used their CNN in dedicated populations with or without cardiac gating. The here presented approach is a combination that works on both, as also published by van Velzen et al. (this work is not cited or discussed by the authors). Despite the application to large cohort studies, there is a lack of novelty in this approach. However, I agree with the authors that their work is overall a step in the right direction.

We would like to thank the reviewer for agreeing with us and acknowledging the importance of our work. Although van Velzen et al. has shown the automation of CAC measurements compared to manual assessments in several clinical scenarios, they did not take the next step to ascertain whether the automated CAC measurements predicted subsequent cardiovascular events, an important requirement before clinical application can be considered. We added this study to the discussion of the revised manuscript (Page 8; Line 217):

A combined solution capable of analyzing cardiac and non-gated chest CT as presented in our investigation has only been proposed by de Vos et al.³⁶ and van Velzen et al.⁴¹. [...] Van Velzen et al. has shown the automation of CAC measurements compared to manual assessments in several clinical scenarios, using a two step approach to find calcification candidates and subsequently detecting calcifications, again using smaller testing cohorts compared to our present study.

- B) The prognostic implications of coronary calcium quantification on prognosis are well known, especially for most of the used cohorts in this study. The quantification by using CNN could definitively support clinical value, but the results to predict outcome is already well documented by the authors and others, as cited in the manuscript.

We agree that the link between coronary artery calcium and cardiovascular events is well documented. Most previous reports of automated coronary artery calcium quantification have focused on accuracy as

compared to humans. We do feel that it is important to take this final step to show that automatic deep-learning derived calcium scores directly predict events, before clinical implementation.

Minor:

- C) The scoring system is demonstrated in US populations, validation in European or Asian cohorts is not existing.

We would like to thank the referee for this important observation. Validation in European, Asian, and other cohorts would be very desirable, but we do not have access to these data. This was one of the reasons that motivated us to make the complete code and trained model publicly available at publication of the manuscript, enabling international researchers to apply and validate our model using their own data. This point was clarified in the discussion (Page 9; Line 244):

The majority of FHS (100%)⁴², NLST (91%)¹⁸, PROMISE (77%)⁴³ and ROMICAT-II (66%)⁴⁴ participants were non-Hispanic whites. Although the manual calcium score has proven to be an important predictor of cardiovascular events across race and ethnicities, generalizability to other demographics will need to be investigated in future studies⁴⁵.

- D) Comparable to manual calcium scoring, this approach will miss all patients without calcified plaques, e.g. patients with vulnerable soft plaques. Quantitative analysis of plaques and stenosis (not only calcifications) by CNN would address this need.

We agree that noncalcified plaque cannot be detected on noncontrast CT. We have added a sentence to the limitations in the discussion stating that *“Furthermore, the proposed system evaluated the coronary artery calcium score on non-contrast cardiac and chest CT. As such it could not detect noncalcified plaque, which can be present even with a calcium score of zero.”* (Page 9; Line 248)

- E) The authors are not demonstrating equality / superiority of outcome prediction compared to manual measurements.

We would like to thank the referee for this remark. We added a comparison of the automatic and manual prognosis results to the manuscript and show detailed results in Table S5. There was no significant difference between manual and automatic AUC for event prediction (see results section *Comparison of automated deep learning to manual coronary artery calcium scoring and event prediction* (Page 6; Line 154), as well as **Table S5**; Supplements). As the network was trained on manual segmentations, we did not expect the network to outperform these manual measurements.

Comparison of automated deep learning to manual coronary artery calcium scoring and event prediction.

[...]

*To show the predictive value of the automatically calculated calcium score we computed the AUCs for event prediction in NLST, PROMISE and ROMICAT-II (**Table S2**) and compared them to AUCs from manually derived calcium scores (**Table S5**). We used random effects meta-analysis to estimate combined predicted and manual AUCs. The combined predicted AUC=0.74 was statistically not different to the combined manual AUC=0.75 (P=0.544).*

Reviewers' Comments:

Reviewer #1:

Remarks to the Author:

My concerns have been addressed.

Reviewer #2:

Remarks to the Author:

The authors present a revised version of their manuscript "Deep convolutional neural networks to predict cardiovascular risk from computed tomography".

The authors have revised the manuscript extensively and addressed some of my concerns. I agree with the authors, that an automated "end-to-end" solution for CAC scoring, that is validated on a broad dataset, demonstrated outcome prediction and is publicly available, will be a valuable step to translate CAC scoring into a broad clinical application.

However, my major critiques in regard to novelty still remain. First, it does not add novelty to demonstrate comparable results in larger (and better characterized) cohorts, if other authors have created sufficient evidence on other "large" scale datasets before. Second, making something available after publication, cannot be regarded as the a scientific novelty of a study (but I think this is highly valuable and want to congratulate the authors of this initiative). Third, others (and the authors) demonstrated the accuracy of deep-learning based CAC-scoring in comparison to manual measurements. Therefore, the comparability in regard to prognostic implications are not surprising nor really novel and just a "last step", even other groups have not demonstrated this before.

Point-by-point response to the questions raised by the reviewers

Reviewer #1 (Remarks to the Author):

My concerns have been addressed.

We would like to thank the referee for this positive assessment.

Reviewer #2 (Remarks to the Author):

The authors present a revised version of their manuscript “Deep convolutional neural networks to predict cardiovascular risk from computed tomography”.

The authors have revised the manuscript extensively and addressed some of my concerns. I agree with the authors that an automated “end-to-end” solution for CAC scoring, that is validated on a broad dataset, demonstrated outcome prediction and is publicly available, will be a valuable step to translate CAC scoring into a broad clinical application.

However, my major critiques in regard to novelty still remain. First, it does not add novelty to demonstrate comparable results in larger (and better characterized) cohorts, if other authors have created sufficient evidence on other “large” scale datasets before. Second, making something available after publication, cannot be regarded as a scientific novelty of a study (but I think this is highly valuable and want to congratulate the authors of this initiative). Third, others (and the authors) demonstrated the accuracy of deep-learning based CAC-scoring in comparison to manual measurements. Therefore, the comparability in regard to prognostic implications are not surprising nor really novel and just a “last step”, even other groups have not demonstrated this before.

First, we agree with the reviewer that our validation of an automated end-to-end CAC scoring solution has value and will help translate CAC scoring into a broad clinical application. We respectfully disagree that this does not add novelty - the novelty is that we have proven that the model works in a diverse cohort of over 20,000 individuals from multiple trials, multiple clinical settings (asymptomatic screening, stable chest pain, acute chest pain), >200 sites, and for multiple adverse cardiovascular outcomes. This speaks to the robustness of our methods and applicability across multiple clinical settings.

Second, this will be the first publicly available open source deep learning model to automatically measure CAC. Free open source distribution is critical for this field, and will allow other investigators to use the model on their datasets and serve as the benchmark for future research. To our knowledge, all previous automated CACS tools have not been open sourced. It is important to mention that our system will be published with the open MIT license and will be released before publication of this manuscript.

Third, we respectfully disagree about the value of going the “last mile” to demonstrate that automated CAC scores predict major adverse cardiac events. These health outcomes are what matter most from a clinical perspective. Proving that our model both accurately quantifies CAC and predicts cardiac events is the gold standard before it should be accepted in clinical practice.